# Applying a Fast-Scan Cyclic Voltammetry to Explore Dopamine Dynamics in Animal Models of Neuropsychiatric Disorders

**DOI:** 10.3390/cells11091533

**Published:** 2022-05-03

**Authors:** Vladimir P. Grinevich, Amir N. Zakirov, Uliana V. Berseneva, Elena V. Gerasimova, Raul R. Gainetdinov, Evgeny A. Budygin

**Affiliations:** 1Department of Neurobiology, Sirius University, 1 Olympic Ave., Sirius, Sochi 353340, Russia; grinevich.vp@talantiuspeh.ru (V.P.G.); zakirovamir@gmail.com (A.N.Z.); berseneva.uv@talantiuspeh.ru (U.V.B.); gerasimova.ev@talantiuspeh.ru (E.V.G.); gaynetdinov.rr@talantiuspeh.ru (R.R.G.); 2Institute of Translational Biomedicine and St. Petersburg State University Hospital, St. Petersburg State University, Universitetskaya Emb. 7-9, St. Petersburg 199034, Russia

**Keywords:** dopamine, neurotransmitter release, fast-scan cyclic voltammetry, addiction, Parkinson’s disease, schizophrenia

## Abstract

Progress in the development of technologies for the real-time monitoring of neurotransmitter dynamics has provided researchers with effective tools for the exploration of etiology and molecular mechanisms of neuropsychiatric disorders. One of these powerful tools is fast-scan cyclic voltammetry (FSCV), a technique which has progressively been used in animal models of diverse pathological conditions associated with alterations in dopamine transmission. Indeed, for several decades FSCV studies have provided substantial insights into our understanding of the role of abnormal dopaminergic transmission in pathogenetic mechanisms of drug and alcohol addiction, Parkinson’s disease, schizophrenia, etc. Here we review the applications of FSCV to research neuropsychiatric disorders with particular attention to recent technological advances.

## 1. Introduction

Neuropsychiatric disorders such as drug or alcohol dependence, Parkinson’s disease, and schizophrenia have undergone decades of research, but their prevalence has not decreased. Mechanisms contributing to these disorders have yet to be fully delineated and this is still an area of ongoing research. Despite constantly enhancing earlier diagnostics, modern health care is still experiencing a deficiency of effective pharmacotherapies. Undoubtedly, the success in developing effective treatments depends on detailed understanding of the neurobiological mechanisms that trigger and promote these pathological conditions. Since dopamine neurotransmission plays a pivotal role in pathogenesis of diseases such as Parkinson’s, schizophrenia, drug abuse, and others, remarkable progress has been made toward understanding how aberrations in dopaminergic transmission impact these pathologies.

In fact, development of pathology relevant animal models in parallel with the advance in monitoring of dynamic changes of neurotransmitters in the brain have provided researchers with a powerful toolbox for investigation in the neuroscience field. One of the most productive methodologies, which has been successfully used for many years, has become FSCV. FSCV was developed and popularized as a revolutionary technique to measure rapid changes of neurotransmitters at a biological time scale by Wightman and Millar in the 1980s as a logical continuation of the extraordinary efforts of the Adams’ group, which pioneered the application of electrochemical methods to quantify in situ endogenous catecholamines [1,2,3,4]. 

The general principle of electrochemical techniques, including FSCV, is based on oxidation and reduction of electroactive compounds when a certain potential is applied to the electrode, and measuring the resulting current. Differences in the ways in which the potential holds and generated current is measured provide benefits and limitations of certain electrochemical methods. The detailed comparison of technical aspects of FSCV with other commonly used approaches such as amperometry, high-speed chronoamperometry, and microdialysis has been provided in previous reviews [5,6,7,8,9,10]. Here, we briefly emphasize the features which make FSCV suitable and perhaps more prevalent than other techniques for the exploration of abnormalities associated with dopamine transmission, while also considering disadvantages.

FSCV and high-speed chronoamperometry apply waveforms, triangle and square, and sample with the resolutions of hundreds of milliseconds, while amperometry holds constant potential, measuring a current at least an order of magnitude faster. Therefore, amperometry offers much better temporal resolution than other real-time electrochemical approaches including FSCV. However, the resolution of FSCV is still sufficient for monitoring fast fluctuation in extracellular dopamine, including naturally-evoked phasic dopamine effluxes observed in different behavioral paradigms on freely moving animals [11,12,13,14,15,16,17,18,19,20,21].

Importantly, in contrast to FSCV and high-speed chronoamperometric measures, amperometry cannot provide information regarding the detecting analyte, since the current is measuring at fixed potential. The negative step in high-speed chronoamperometric measures allows this technique to establish a proportion between reductive and oxidative current that helps with analyte identification. FSCV shares with high-speed chronoamperometry this critical advantage in regard to a chemical resolution. Moreover, the use of background-subtracted voltammograms, which provide distinguishing current profiles within different potentials, affords FSCV an upgraded capability to reveal the identity of the detected substance. In contrast to amperometry, FSCV is sensitive to pH alterations, which are often associated with neuronal activation and therefore can contaminate the detecting signal. Nevertheless, based on the voltammogram, FSCV is capable to separate dopamine from pH changes. The special method was developed to statistically distinguish dopamine spikes using principal component regression [18]. This approach is especially valuable for revealing dopamine transients collected in freely moving animals, providing the exceptional opportunity to link the neurotransmitter release with behavior. Remarkably, a machine learning method was offered recently to uncover dopamine spikes in FSCV recordings performed in the human brain [22]. 

FSCV coupled to electrical stimulation is one of the few approaches comprehensively used to evaluate functional uptake of endogenous dopamine in brain slices and anesthetized preparations [5,23,24,25,26,27,28,29,30,31]. Since the method measures extracellular dopamine concentrations at a subsecond time scale inducing minimal tissue damage due to the micrometer dimension of electrodes, changes in dopamine uptake can be sensibly determined in small brain subregions. The signal can be analyzed by Michaelis–Menten model, providing information on dopamine concentration released per stimulation pulse and separating changes in Vmax and Km parameters, which characterized different aspects of dopamine uptake kinetics. This was particularly beneficial to explore neuroadaptations in dopamine transmission in models of drug addiction [23,24,25,26,27,28] and Parkinson’s disease [32].

By comparison, intracerebral microdialysis coupled with HPLC measurements, a technique with better chemical selectivity and sensitivity, can reliably determine basal extracellular concentrations of neurotransmitters. However, microdialysis probes are much larger than FSCV electrodes and thus create more damage to the brain tissue, and this approach does not have the advantage to measure detached release and uptake changes due to poor temporal resolution. Thus, microdialysis and FSCV focus on different aspects of neurotransmitter dynamics and these approaches have been widely used in parallel for decades [33,34,35,36,37,38,39]. Overall, among the electrochemical methods, FSCV coupled with sophisticated kinetic models is the most useful technique for exploring the details of dopamine release and uptake changes both in vitro and in vivo (Figure 1), including in freely moving animals. Alone and in combination with other up-to-date approaches and behavioral models, FSCV has prominently contributed to our understanding of the dopamine role in psychiatric and neurodegenerative diseases. Certainly, FSCV is not the only method to benefit the neurobiology field. Nowadays, biosensors permit the examination of chemical neurotransmission with millisecond temporal and single cell spatial resolutions. Thus, recently developed fluorescent biosensors for photometric detection of dopamine in vivo may open breakthrough opportunities to provide further insight into neurochemical mechanisms of neuropsychiatric disorders [40]. It should be noted, however, that the use of biosensors requires complex genetic manipulations involving expression of proteins in the animal brain using adeno-associated viral vectors and the use of Cre recombinase expressing animals, complicating reproducibility and routine use of these techniques by the majority of laboratories. Nevertheless, this critical review is shedding light for the first time on significant findings for the neuropsychiatry topics, which have become possible due to advantages of FSCV over other closely related technologies.

## 2. Drug and Alcohol Addictions

Drug and alcohol addictions, also called substance use disorders (SUDs), are chronic psychiatric conditions involving complex interactions between brain neurotransmitter systems, genetics, and environment. SUDs are characterized by uncontrolled substance seeking and taking behaviors, and long-lasting vulnerability to relapse. The key features of these neuropsychiatric pathologies can be reproduced in animal models including different paradigms of substance self-administration in rodents. Such models are widely used for the exploration of the neurobiological mechanisms underlying the development and escalation of addictions. A wealth of evidence demonstrated that altered dopamine signaling is involved in all stages of addiction, from initiation and maintenance to escalation and relapse [41]. Due to a high temporal resolution and sufficient chemical specificity, FCSV was especially helpful in evaluating the details of fast dopamine transmission changes in different experimental paradigms. 

Several critical factors were implicated for the development, shaping, and escalation of addictive behaviors. The first of them is a prominent brain response triggered by stimuli, which were previously associated with the exposure to abused substances. This response can be tightly linked with a desire and motivation to obtain a drug, while a learning function was equally proposed as well. The discovery that projections from the ventral tegmental area (VTA) to the nucleus accumbens (NAc) release dopamine at subsecond time scale during the presentation of cocaine-related cues in rats trained to self-administer the drug was made with the use of FSCV [11]. Moreover, rapid changes in extracellular dopamine concentrations were detected in the NAc of rats before lever presses for cocaine in coincidence with the initiation of drug-seeking behaviors. A subsequent study revealed that extinction of cocaine self-administration resulted in functionally and temporally distinct dopamine signals in the same brain region [12]. Specifically, phasic dopamine release that occurs following a lever-press for cocaine gradually diminishes during extinction, but the signal that arises before the pressing episode appears persistent to extinction. Moreover, drug-seeking behaviors could be reproduced by evoking accumbal dopamine release by electrical stimulation of the VTA [11]. These findings suggest that subsecond dopamine efflux (phasic) may promote drug seeking if, indeed, a cause-and-effect relationship between the neurochemical changes and induced behavior exists. However, since the consequence of electrical VTA stimulation can be the release of various neurotransmitters in different brain areas, including dopamine [42], a causal role of the release of this neurotransmitter in the nucleus accumbens in the triggering or enhancing of drug-seeking (motivational) behavior can be questioned. 

Other support for a possible role of phasic dopamine in promoting motivated behavior comes from voltammetric studies demonstrating that repeated cocaine exposure may increase cue-evoked motivation via augmented phasic dopamine release [13]. Furthermore, phasic dopamine changes were positively correlated with lever pressing for reward, while slow alterations in dopamine concentration (tonic) were not linked to this activity [14]. Although these studies advocate for the role of accumbal dopamine in reward-seeking behaviors, this evidence remains correlative in nature. 

An alternative hypothesis postulates that phasic dopamine signaling may serve a teaching function, encoding the association between cues that predicted reward events and report errors in reward prediction. In fact, these postulates are based on compelling data [15,16,17,43,44] including the proof of the causal link between dopamine and learning from optogenetic studies [45]. Perhaps these two hypotheses can coexist without opposing each other, since the learning component is causally involved in cue-elicited reward-seeking through a phasic dopamine signaling [45]. Nevertheless, there is the need for additional studies, which should combine optogenetic approaches and FSCV recordings in a cocaine self-administration paradigm, in order to prove the causality between phasic dopamine release and triggering drug-seeking behaviors independent from the learning. 

On the other hand, some progress was made in the understanding of the causative relationship between dopamine release and alcohol-seeking behavior. Using FSCV recordings coupled with the selective optogenetic activation of dopamine VTA-nucleus accumbens circuitry, it was confirmed that high and low frequencies may generate phasic and tonic increases in dopamine release, respectively [46,47], which were previously detected in behaving animals [11,12,13,48]. This allowed Budygin’s group to explore the relationship between dopamine transmission patterns and alcohol self-administration [49]. It has been revealed that phasic pattern of accumbal dopamine transmission within mesolimbic circuitry enhanced alcohol seeking, whereas tonic pattern inhibited alcohol-seeking and taking behaviors [47,50]. The fact that these two patterns had opposite effects on alcohol-seeking behavior emphasizes the importance of temporal dopamine dynamics in controlling motivated behaviors. To understand the contrary behavioral consequences induced by tonic and phasic dopamine increases, optogenetics and FSCV methods were integrated in anesthetized rats [47]. This approach allowed the researchers to directly explore real-time interaction between two distinct patterns of dopamine transmission. Therefore, it was found that simultaneous generation of tonic and phasic increases resulted in an inhibition of the large dopamine response observed with phasic stimulation alone. Consequently, sustained tonic stimulation of accumbal dopamine may decrease alcohol seeking by preventing dopamine terminals from engaging in phasic signaling patterns that promote alcohol-seeking behavior under normal circumstances. Together, these results implicate dopamine release in the ventral striatum as a critical neurochemical substrate, controlling behaviors directed to obtain abused substance and emphasize the role of discrete patterns of the neurotransmission for the initiation and inhibition of this action.

A second factor that may play a key role in the escalation of addiction as well as in relapse is the neuroadaptation induced by chronic use of a drug. One of the important consequences of repeated increases in extracellular dopamine concentrations following prolonged exposure of addictive substances, including cocaine and alcohol, is an altered reuptake rate. FSCV coupled with electrical stimulation offers some advantages for the evaluation of these changes. Since the technique detects dopamine changes with subsecond resolution at micrometer-dimension probe, endogenous reuptake can be measured in real time with minimal tissue damage. Moreover, FSCV is able to resolve changes in dopamine uptake kinetically, separating changes in maximal uptake rate (Vmax) and Km parameters. The measurements can be performed in vitro, using brain slice preparations, and in vivo on anesthetized and freely moving animals.

Due to the above-mentioned advantages of FSCV, the consequences of escalated cocaine self-administration (0.75 mg/kg, fixed-ratio, 6 h session for 14 days) on dopamine uptake in rat nucleus accumbens were revealed. First of all, it was found that cocaine-induced dopamine uptake inhibition (changes in Km) reached a proportionally higher level during the loading phase of cocaine self-administration [23]. Correspondingly, the dopamine uptake inhibition thresholds associated with the maintenance of responding were rising with new compressed intervals between self-injections. These changes were consistent with an escalation of cocaine-taking behaviors. Examination of dopamine uptake rate parameters revealed that Vmax is significantly increased following long-access escalation training, while no changes in the affinity (Km) of cocaine for the dopamine transporter were observed [23]. Using a self-administration paradigm that did not result to escalated intake of cocaine, no alterations in either uptake parameter were found in another in vivo voltammetry study [24]. However, in rats self-injecting the combination of cocaine and heroin (speedball), there was a significant increase in the Vmax. The combined effect of cocaine and opiates on dopamine transmission is of special interest when considering the unique neurochemical profiles reported with speedball that are thought to contribute to the increased abuse potential. Microdialysis studies have shown that speedball induces a synergistic elevation in extracellular dopamine concentrations in the nucleus accumbens compared to cocaine or heroin alone [51]. Therefore, the enhanced rate of dopamine uptake (Vmax) is likely the result of an upregulation in functional dopamine transporter efficiency in response to the increased dopamine level. 

In parallel with in vivo findings, the rate of accumbal dopamine uptake (Vmax) measured with FSCV in brain slice preparation ex vivo was significantly increased following a 10-day binge of cocaine self-administration (1.5 mg/kg, fixed-ratio for 24 h) [25]. However, a 5-day cocaine self-administration paradigm (1.5 mg/kg, 40 injection per day) was not sufficient to induce Vmax adaptation in another ex vivo voltammetry study [26]. Therefore, this presynaptic neuroadaptation may occur following relatively prolonged cocaine exposure, probably as a compensatory response of the system to persistently elevated extrasynaptic dopamine concentration.

Ex vivo FSCV was used to explore the consequences of chronic alcohol exposure on dopamine transporter-mediated uptake parameters. Thus, the acceleration in striatal dopamine uptake (Vmax) was observed in brain slices obtained from monkeys that were taking alcohol over a year [27], and in rats [28,29] and mice [30] chronically exposed to ethanol vapor or solution. It has been shown that chronic ethanol treatment leads to an increase in dopamine transporter (DAT) levels [52]. Consequently, an enhanced reuptake can be involved in the decrease of extracellular dopamine concentrations following chronic ethanol exposure. 

Considering DAT as the initial target, which is responsible for cocaine’s reinforcing action, the most intriguing results obtained with FSCV regarding neuroadaptations of presynaptic dopamine function are findings that DAT becomes less sensitive to cocaine inhibition following binge-like self-administration procedure [25,31]. Analysis of dopamine kinetics in brain slices revealed that cocaine was markedly less effective in inhibiting the uptake (apparent Km) [25]. Noticeably, there were similar changes in Km in rats at 1 and 7 days of cocaine deprivation. This is in sharp contrast to behavioral experiments, which have shown that the reinforcing effects of the drug are significantly augmented after 7 days of deprivation [25,53]. Therefore, adaptive alterations due to changes of the cocaine efficacy cannot account for distinct behavioral consequences following a binge/deprivation history of self-administration. However, these changes may be involved in a key mechanism underlying self-reports of lessened euphoria or tolerance produced by psychostimulants in addicts [26]. Therefore, decreased sensitivity of the DAT to cocaine could be associated with development of the tolerance. 

Due to its excellent spatial resolution (~6 µm × 100 µm), FSCV is capable of accurate monitoring of dopamine release changes in distinct brain subregions of freely moving animals. This advantage allowed Phillips’ group to discover hierarchical recruitment of phasic dopamine signaling in the striatum during the progression of cocaine self-administration. They carried out longitudinal subsecond dopamine detections simultaneously in the ventromedial striatum (VMS) and dorsolateral striatum (DLS) during the establishment of drug taking in rats [19]. They detected dopamine releases in both the VMS and DLS following the operant response for cocaine during the course of self-administration, in which the VMS signal declined and the DLS signal emerged during the progression of drug taking. The finding of these neuroadaptations within limbic and sensorimotor striatal projections provides insight into neurobiological processes that establish drug-taking habits. Therefore, the progression of drug taking beyond recreational use might indeed reflect the engagement of dopamine signaling in distinct areas of striatum [54,55], with an emphasis of shift from the VMS to DLS during the development of established drug-seeking behavior [54,56].

The consequent question answered by the same researchers [20] was whether drug-directed behaviors are encoded by phasic dopamine release as drug taking escalates. By combining FSCV with self-administration regimen that is capable of producing escalated and compulsive drug seeking, they explored the regional dynamics of dopamine signaling following long access to cocaine. Remarkably, it was found that phasic dopamine decreased in both VMS and DLS regions as the rate of cocaine intake increased; with the reduction in neurotransmitter in the VMS significantly correlated with rate of escalation [20]. Furthermore, L-DOPA at a dose that refilled dopamine release reversed escalation, thus demonstrating the causal relationship between decreased dopamine transmission in the VMS and escalated cocaine self-administration [20]. Results from this voltammetric study provided mechanistic as well as therapeutic insights into excessive drug-taking behavior.

Stress is the third factor with a well-established role in escalation and relapse of drug- seeking and -taking behaviors in SUDs that can be reproduced in the animal models. However, dopaminergic mechanisms by which stress impacts addictive behaviors are still unclear. The evolutionary significance of the brain response to stress is to orchestrate adaptations, which have been influenced by natural selection to mobilize the individual bodily abilities to optimally deal with situations that needs escaping action or defense (flight or fight). The increase in phasic dopamine detected with FSCV in the NAc of intruder rat during social defeat episodes [21] is perhaps a part of this response. Consequent voltammetric studies in vitro revealed that stress-induced dopamine release is triggered by corticotropin-releasing factor (CRF) [57,58,59] that orchestrates brain responsiveness to stress exposure. One remarkable finding is that CRF loses the ability to increase mesolimbic dopamine release [57,58,59] and produces an aversive behavioral response [57]. It should be highlighted that experience-dependent dysregulation of the CRF system has been considered as a main contributor to vulnerability for stress hyperresponsivity as well as addictive behaviors. Collectively, these voltammetric findings pointed in a promising direction for the identification of neurochemical mechanisms underlying stress-promoted (associated) development and escalation of addictive behaviors, that is, the CRF-dopamine release interaction in the nucleus accumbens.

## 3. Parkinson’s Disease

Parkinson’s disease (PD) is a progressive neurodegenerative disorder that mainly affects the motor system with clinical manifestations occurring after the pathology reaches its advanced stage [60,61,62]. The dominant motor symptoms of the disease such as resting tremor, rigidity, and bradykinesia manifest the loss of the dopamine neurons within the substantia nigra part compacta, which project to the caudate nucleus and posterior putamen [63,64]. PD motor symptoms appear gradually as cells in the substantia nigra die and striatal dopamine levels drop [65,66].

Although PD has been intensively studied from a variety of aspects, there is no effective cure for PD, largely due to a lack of mechanistic understanding of its pathogenesis. It is still unclear whether a progressive degeneration of dopaminergic neurons is the sole causative mechanism occurring before clinical manifestation of PD, or whether there are concomitant losses in other neuronal systems, including the cholinergic system. The latter is known as also being progressively affected by the PD degenerative process [67,68]. The numerous recent findings suggest that aberrations in the cholinergic system and nicotinic acetylcholine receptors (nAChRs) contribute to the pathogenetic mechanisms of PD concomitantly with the losses in the nigrostriatal system [69,70,71,72]. It is important to note that heteromeric β2 subunit-containing nAChRs, in particular, are expressed on dopamine terminal in the striatum [73,74,75] and mediate accompanying changes in the tone of the cholinergic interneurons (CIs), which reside in short proximity to the dopamine terminals in the striatum [76].

To explore a direct effect the presynaptic β2-containing nAChRs produce on dopamine release in the striatum during different patterns of nigrostriatal dopamine signaling, series of ex vivo FSCV studies utilizing rat and guinea pig brain slices were conducted [77,78,79]. Based on results from pharmacological approach, these β2-containing nAChRs, expressed presynaptically on dopamine axon terminals, were implicated to modulate translation of dopamine neuron action potential into dopamine release. Specifically, a full nAChRs agonist nicotine was used in comparison with selective β2-nAChRs antagonist dihydro-β-erythroidine or non-selective allosteric nAChRs antagonist mecamylamine to assess possible mechanisms underlying nAChRs-mediated modulation of dopamine neurons [77,78]. It was observed that nicotine acts bidirectionally on electrically evoked dopamine release depending on number of pulses and frequency of locally applied stimuli: it suppressed dopamine release evoked by single pulse or by pulses of low frequency but proportionally facilitated dopamine release evoked by high frequency stimulation (the higher frequency the higher facilitation). The fact of suppression of the basal dopamine tone indicates that endogenous acetylcholine most likely activates presynaptic nAChRs, which in turn slightly stimulate dopamine release in striatal terminals. Moreover, nicotine produced analogous effect to those elicited by both antagonists suggesting desensitization of nAChRs as a mechanism of its action. The enhancement of phasic-to-tonic dopamine release in the striatal terminals was also observed while acetylcholine synthesis was selectively eliminated in the forebrain utilizing the choline acetyltransferase forebrain knock-out model in mice [80]. Thus, pauses in CIs tone may permit more efficient translation of burst-firing action potentials into phasic dopamine release, enhancing signal gain at axon terminals. The combination of electrophysiological recordings of striatal channelrhodopsin2-expressing cholinergic interneurons with simultaneous FSCV detection of dopamine release in striatal slices indicated that synchronized action potentials within a local CIs-dopamine network activated nAChRs and enhanced axonal dopamine release in ex vivo slices and in vivo [81,82] bypassing action potential generation in the midbrain. Dopamine release can be driven by nAChRs in response to direct CIs activation or indirect CIs activation by optogenetic stimulation of glutamatergic inputs, e.g., from the thalamus [81].

In summary, so far, striatal dopamine slice FSCV demonstrated that activity in dopamine cell bodies is not an exclusive trigger for terminal dopamine release; striatal ACh acting at nAChRs on dopamine axons bypasses signal propagation in midbrain dopamine neurons by triggering terminal dopamine release directly. Thus, nAChR-active compounds may be implicated at least as adjunct therapies for PD, but more comprehensive studies are needed.

FSCV was instrumental in uncovering alterations in dopamine neuronal functions in several transgenic animal models relevant to PD. Several studies were devoted to understanding the role of presynaptic protein alpha-synuclein in dopaminergic neurotransmission in the nigrostriatal dopaminergic system. Mutations in this protein are known to be linked to the pathogenesis of PD. Deficient evoked striatal dopamine release was documented in transgenic mice overexpressing mutated human A30P alpha-synuclein under the control of the promoter of prion-related protein [83], bacterial artificial chromosome (BAC) transgenic mice expressing human unmutated alpha-synuclein at disease-relevant levels [84], and in BAC transgenic mice that were created by the expression of the mutated human A30P alpha-synuclein in knockout mice lacking alpha-synuclein [85]. In contrast, enhanced facilitation of dopamine overflow in dorsal striatum following repeated, high-frequency burst stimulation of the dopaminergic pathways, likely due to modification of dopamine vesicles found in mice lacking alpha-synuclein [86], and elevated evoked release of dopamine in the dorsal striatum was documented in mice lacking all synucleins (alpha, beta, and gamma) [87]. Furthermore, in two alpha-synuclein deficient mouse lines, one carrying a spontaneous deletion of alpha-synuclein locus and the other a transgenic alpha-synuclein knockout, stimulated dopamine overflow was higher with a concomitant decrease in dopamine reuptake in the dorsal striatum [86]. Furthermore, human alpha-synuclein-overexpressing transgenic mice, a model of early PD, demonstrate elevated dopamine uptake rates and more pronounced effects of DAT inhibitors on evoked extracellular dopamine concentrations [88]. Utilizing ex vivo slice FSCV it has been demonstrated that alpha-synuclein promotes striatal DAT function through mechanisms dependent on extracellular cholesterol, thus suggesting converging biology of alpha-synuclein and cholesterol to regulate DAT that potentially promotes vulnerability to neuronal degeneration in PD [88]. In a human alpha-synuclein-expressing mouse model of PD with macroautophagic failure in dopamine neurons, an increase in evoked extracellular concentrations of dopamine, reduction of dopamine uptake, and relieved paired-stimulus depression was found [89]. At the same time, in mouse model of enhanced vesicular function via BAC-mediated overexpression of the vesicular monoamine transporter 2 (VMAT2; Slc18a2) a 2-fold elevated vesicular capacity led to an increase in stimulated dopamine release [90]. The extended pre-symptomatic phase of PD, when its core motor symptoms do not appear before degeneration of nigrostriatal dopaminergic neurons exceed their critical level (50–70%) and striatal dopamine concentration does not fall below 60–70%, imply a vital role for associated compensatory mechanisms to effectively maintain motor functions [91,92,93,94]. Thus, it is of importance to focus on elucidation and effective management of mechanisms that are critical for the maintenance of dopaminergic activity up to the end of prodromal motor periods [95]. To investigate the mechanistic plasticity of dopamine release, in vivo voltammetry was used in studies on the parkinsonian rats, which underwent an electrical stimulation protocol producing an exhaustion (“fatigue”) of nigrostriatal dopaminergic neurons [32]. In these studies, Bergstrom et al. used FSCV and applied repetitive, high-frequency stimulation (60 Hz, every 5 min for 60 min) to investigate the effects of dopamine depletion on time-dependent dopamine release. Utilizing the approach mentioned above, authors identified a novel compensatory mechanism for the maintenance of dopamine release in vivo that was independent of activated neurotransmitter synthesis but driven by altered neurotransmitter uptake. Notably, amplitudes of electrochemical signals were maintained in unilateral partially 6-OHDA-lesioned striata in response to a fatiguing protocol that significantly decreased evoked signal levels in intact striata. The sustained release was replicated in intact animals after blockade of DAT via an irreversible DAT inhibitor RTI-76 [96], implicating neuronal uptake as a trigger. While their results supported the close association between synthesis and release [97], perhaps because newly synthesized dopamine preferentially releases [98], authors suggested that activation of dopamine synthesis is not necessary for its release to adapt in those conditions. This newly described neuroadaptation may contribute to early preclinical normalization of function and help resolve discrepant findings regarding compensatory changes in dopamine release during the progression of the parkinsonian state. In contrast to these time-dependent changes in dopamine release, a different result emerged when the interval between stimulus trains was sufficient to permit a return of the factors controlling release to a stable baseline. In the latter case, the release was constant over time or, as such, time-independent [99,100]. When dopamine release in 6-OHDA-lesioned rats was studied under time-independent conditions, it was found that dopamine concentration in the extrasynaptic space was decreased proportionally to a degree of progressed denervation [101,102]. However, dopamine release, observed under time-dependent conditions in the same 6-OHDA-lesioned rats, remained persistent regardless of a degree of neuronal loss [32]. 

Thus, an implication of FSCV into studies on mechanistic understanding of the pathogenesis of PD allowed for identification of novel compensatory maintenance of the striatal dopamine release suggestive of a functional reorganization of the presynaptic dopaminergic terminals in the early parkinsonian state. 

It is well-known that neurons of the dopaminergic system communicate under two modes, a fast “phasic” release (subsecond to second) and a slower “tonic” release (minutes to hours). Besides others, alterations in tonic dopamine levels are believed to be more critically important in enabling normal motor functioning, and dysregulation in tonic dopamine levels is believed to be causative to the development and progress of movement disabilities, which are crucial for PD. Therefore, the employment of neurochemical techniques that enable rapid, selective, and quantitative measurements of changes in tonic extracellular levels are essential in determining the role of dopamine in both normal and disease states. At present, electrochemical techniques for tonic dopamine concentration measurements notably encompass FSCV as one of the most preferred techniques due to its high spatial, temporal, and chemical resolution, as well as because of its great promise for the studies on disease neuropathogenesis in humans. Additional modified FSCV techniques have also been developed and become instrumental for measurements of tonic dopamine by the present time [103,104,105]. Such modified FSCV techniques demonstrate consistency by providing similar extracellular dopamine concentration at axon terminal areas when used in different in vivo animal models. Besides modified FSCV techniques, a number of complex voltammetric waveform methods have been designed and applied [106,107,108]. One particular method utilizes multiple cyclic square waves in combination with dynamic background subtraction and current modeling (multiple-cyclic square wave voltammetry) to retain accuracy and reproducibility in assessing neurotransmitter at its tonic levels [107,109].

Unfortunately, these capabilities of the voltammetry to accurately evaluate changes in basal dopamine concentration are underappreciated so far. In fact, the use of the above-mentioned advantages could further benefit studies aimed to understand the mechanisms dysregulation in tonic dopamine transmission under the development of PD.

For a more in-depth understanding of the advantages of FSCV for studies on dopamine in vivo, please refer to the substantial reviews on voltammetric measurements of brain neurotransmitters by Bucher and Wightman [110] and by Banerjee and co-authors [9].

## 4. Schizophrenia

Schizophrenia is a complex devastating psychiatric disease of which pathogenesis has been debated for decades. It affects about 1% of the population worldwide and is manifested mainly by continuous or relapsing episodes of psychosis. Schizophrenia symptoms fall into three major categories: positive symptoms (visual and auditory hallucinations, various manifestations of delusions, paranoia, and disorganized thinking); negative symptoms (anhedonia, social withdrawal, decreased emotionality, alogia, and avolition); and cognitive deficits (deficient memory, impaired attention, and reduced executive functions) [111]. Following the discovery of the antipsychotic action of chlorpromazine more than 60 years ago, many antipsychotics have been developed, but essentially all of them share a common mechanism of action involving blockade of dopamine D2 receptor. A newer generation of antipsychotics, termed as atypical or second-generation antipsychotics, can also block 5HT2A serotonin receptors and have a somewhat better safety profile, but the efficacy is not much improved over typical antipsychotics and is mostly related to positive symptoms [112,113]. Further limitations in the clinical use of current antipsychotics include prominent side effects, such as extrapyramidal dysfunctions and metabolic dysregulations [114]. Several hypotheses have been proposed to explain the nature of schizophrenia. The central theory emphasizes aberrations in the dopaminergic and glutamatergic systems; nevertheless, numerous other studies suggest the involvement of the cholinergic, serotonergic, GABA-ergic, and even the neurotensin system in the disease [115,116,117,118,119,120,121]. Based on various hypotheses, attempts have been made to create pharmacological and genetic models of schizophrenia in animals. Such models were used to study the disease and to develop medications for treatment of schizophrenia [122,123,124,125,126]. However, it is clear that while these models are useful to investigate certain endophenotypes of schizophrenia, none of them exhibit the full range of symptoms of the disease [127]. Moreover, it remains unclear to what extent these animal models reflect real changes in the brain neurotransmitter systems. One possible approach for studying neurotransmitters in the brain during neurodegenerative diseases in animals is FSCV; it allows quantification of synaptic release of dopamine under in vivo and in vitro/ex vivo conditions.

The dopamine theory is supported by the observation that dopamine hyperactivity in the nigrostriatal and mesolimbic systems or hypersensitivity to dopamine underlies the symptoms of schizophrenia and psychosis [99,119,128,129,130,131]. Dopamine theory is supported by the observation that all clinically used antipsychotics antagonize D2 dopamine receptors that were identified in pharmacological animal models of over-activation of dopamine system, i.e., following amphetamine treatment [132]. Although, recently it was demonstrated that some typical and atypical antipsychotics are potent noncompetitive inhibitors of neuronal nicotinic acetylcholine receptors at concentrations similar to plasma levels achieved in schizophrenia patients [133]. A hyperdopaminergic state has been created in mice [134] and rats [35] genetically by deleting DAT gene. DAT knockout (DAT-KO) animals demonstrated 5–7-fold increased extracellular dopamine due to lack of dopamine re-uptake leading to spontaneous locomotor hyperactivity and cognitive dysfunctions [33,35]. FSCV was instrumental to demonstrate markedly increased extracellular clearance of dopamine following its release that was increased 300-fold in DAT-KO mice [33,134] and about 100-fold in DAT-KO rats [35]. Striatal dopamine clearance rate in DAT-KO mice was not affected by cocaine, amphetamine, MAO, and COMT inhibition and blockade of serotonin or norepinephrine transporters, indicating that only diffusion can regulate extracellular dopamine clearance in the condition of DAT deficiency [33,134]. Essentially, the same observations were gained in FSCV experiments in DAT-KO rats but some minor contribution of MAO to striatal dopamine clearance from the synaptic cleft was revealed [35]. Similarly, a decreased dopamine clearance rate in the striatum was found in DAT knockdown mice expressing 5% of DAT [135] and opposite effect was found in DAT overexpressing mice that have 150% increase in DAT levels [136]. DAT-KO mice and rats are routinely used in pharmacological studies aimed at identification of novel molecular targets for drugs active at hyperdopaminergic states, such as schizophrenia [35,127]. One such investigation led to identification of a new target for the development of antipsychotics. TAAR1 is a G protein-coupled receptor that modulates dopaminergic, serotonergic, and glutamatergic activity in rodents. In fact, TAAR1 agonists effectively inhibit hyperactivity of DAT-KO animals [35,137,138,139]. In clinical trials, the first tested TAAR1 agonist with 5-HT1A agonist activity Ulotaront showed significant efficacy in treating patients with schizophrenia on both positive and negative symptoms without causing the side effects of existing antipsychotics [140]. Ulotaront emerges as the first representative of new generation of antipsychotics not directly affecting D2 dopamine receptor function. To determine whether deletion of the TAAR1 gene or application of TAAR1 ligands disrupts functional presynaptic activity of dopamine neurons, researchers assessed extracellular dopamine dynamics using FSCV in the dorsal striatum and NAc in TAAR1-KO mice [141]. TAAR1 is able to regulate dopamine release predominantly in the NAc, exerting negative modulation of dopamine release. Thus, TAAR1-KO mice have increased evoked dopamine release compared with control mice. No alterations in dopamine clearance, which are mediated by the DAT, were observed. A functional link between TAAR1 and D2R autoreceptors localized at dopamine terminals was shown, confirming that TAAR1 and D2R can modulate each other’s activity. These results contribute to the development of a strategy to correct the symptoms of schizophrenia by modulating dopaminergic system indirectly through activation of TAAR1 [141].

A second major theory postulates glutamate NMDA deficiency in schizophrenia [118,142,143]. In experimental animals this condition is modelled pharmacologically by treatment of rodents with NMDA antagonists that cause hyperactivity as well as deficits in social interaction and cognition [127,132]. Similar schizophrenia-related phenotype was observed in NMDA receptor knockdown mice expressing 5% of glutamate NMDA receptors [144]. These behavioral abnormalities may be caused by dysregulation of glutamate-dopamine interaction [142,143]. To explore how sustained NMDAR hypofunction throughout development affects dopamine system voltammetric studies were performed in these mutants [145]. These studies revealed that dopamine synthesis and release were attenuated, and dopamine clearance was increased in NR1 knockdown mice. At the same time, dopamine D2 autoreceptors were desensitized and dopamine neurons had higher tonic firing rates in mutants. Furthermore, phasic signaling was impaired and dopamine overflow could not be achieved with exogenous high-frequency stimulation that modeled phasic firing. Thorough examination of dopamine neurotransmission involving several approaches including FSCV provided evidence that chronic NMDAR hypofunction leads to a state of elevated synaptic dopamine [145].

Another possibility of glutamate-dopamine interaction may involve mGluR-mediated mechanisms [146]. Pretreatment with a non-competitive NMDA receptor antagonist, phencyclidine (PCP), was hypothesized to disrupt mGluR-mediated modulation of dopamine release in the NAc shell [147]. Using FSCV, it was found that PCP pretreatment resulted in enhanced K+-stimulated dopamine release in NAc brain slices in vitro. Activation of mGluR2 and mGluR5 caused decreased stimulated dopamine release but these effects were independent of PCP action, suggesting that hypersensitivity of dopamine systems in this model is not related to these mGluR [147].

FSCV was particularly instrumental in exploring other potential mechanisms in the pathogenesis of schizophrenia in various other experimental animal models. Administration of a neurotoxin which reduces DNA synthesis methylazoxymethanol to rodents on embryonic day 17 (E17) results in neurophysiological, cognitive, and functional brain changes, exhibiting symptoms of schizophrenia in animals [148,149]. This treatment causes neurochemical dysregulation and changes in prefrontal-cortical and mesolimbic circuits. Studies using FSCV and behavioral tests have shown that the methylazoxymethanol model is able to recapitulate important aspects of schizophrenia such as changes in mesolimbic neurotransmission and abnormalities in reward-oriented behavior [150]. Another potential mechanism of neurochemical dysregulation in schizophrenia and other dopaminergic pathologies might occur due to altered regulation of neurotensin signaling [151]. Neurotensin can modulate dopamine D2 autoreceptor function, thereby affecting dopamine neuronal activity. Indeed, in FSCV study, neurotensin was able to decrease dopamine release [120]. 

Cholinergic regulation of dopamine function plays a key role in many behavioral manifestations [152,153] and dopamine/acetylcholine imbalance can lead to the dopaminergic system malfunction that occurs in several psychiatric disorders, including schizophrenia [115,154,155,156]. To identify the mechanisms underlying this imbalance, FSCV in rat brain slices was used to measure dopamine release induced by high-frequency electrical stimulation mimicking the phase activity of dopamine in NAc by pretreatment with phencyclidine. Subchronic pretreatment with PCP affected the ability of acetylcholine to modulate electrically stimulated dopamine release through the nAChR but had no effect on dopamine reuptake [157]. Variability in CACNA1C gene, which encodes the L-type calcium channel Ca v 1.2, has been linked to the development of schizophrenia and other mental illnesses [158]. There is also evidence that Cacna1c is involved in the regulation of addictive behavior mediated by L-type calcium channel modulation in the mesolimbic dopamine system [159]. It has been proposed that evoked dopamine release and/or reuptake can be modulated by Cacna1c levels. Indeed, FSCV study has revealed that that subsecond dopamine release in the NAc of CACNA1C haploinsufficient mice lacks normal sensitivity to inhibition of the DAT that leads to an impaired behavioral response to dopamine reuptake blockers [160].

## 5. Conclusions

Despite long-lasting comprehensive fundamental and applied studies on neuropsychiatric diseases, modern medicine is still experiencing a deficit of effective pharmacotherapies. This is partly because the field needs better understanding of molecular and neural mechanisms that underlie the pathogenesis of brain diseases. From one side, it can be explained by a difficulty in the development of translational animal models for pathological conditions. From another side, previous achievements were dependent on the contemporary state of neurobiological technologies to analyze molecular mechanisms involved. Most detection systems used, at best, were limited in temporal and spatial resolutions as well as in their ability to conduct real-time analytical approaches on a system level. One successful technique, which is able to overcome those limits, is FSCV. Indeed, FSCV studies in animal models have provided significant insights into the role of altered dopaminergic transmission in neurochemical mechanisms of drug addiction, PD, and schizophrenia. Perhaps, the exploration of other neuropsychiatric pathologies such as attention deficit hyperactivity disorder, Alzheimer’s disease, and Tourette’s syndrome in relation to dopamine abnormalities could also benefit from FSCV. For instance, just a few animal studies on Alzheimer’s disease have reported on utilization of FSCV so far [161,162]. Similarly, there are few explorations of the role of DAT in attention deficit hyperactivity disorder-related animal models [135,163,164].

A new promising application of FSCV has started in pre-clinical and clinical studies. Nowadays, a number of research teams have pioneered in vivo FSCV studies on the human brain and have made encouraging efforts to bring FSCV to patients [165,166,167,168]. However, a good deal of awareness should be taken into consideration by the scientific and clinical communities in assessing the risks of such strategies for patient safety [169].

Finally, FSCV is a constantly and rapidly progressing technique. Recent advances include novel electrode technologies that enable more sensitive, less damaging, long term and wireless recordings, optimized data processing and enhanced analyses. However, future developments demand a combining of FSCV with other leading-edge technologies, such as multisite detection and machine learning [170]. Consequently, further progress in FSCV methodology is essential to facilitate our knowledge of neurochemical mechanisms of brain disorders that could eventually lead to development of more effective therapies.

## Figures and Tables

**Figure 1 cells-11-01533-f001:**
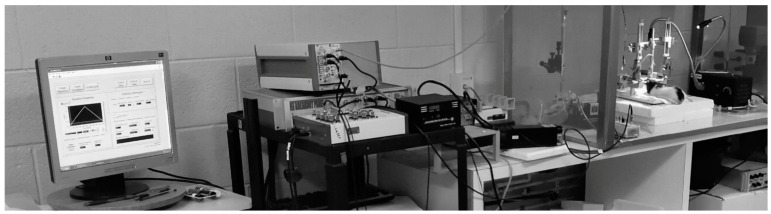
Modern FSCV set up for the in vivo monitoring of extracellular dopamine in the rodent brain. The voltammetry data collection hardware and software were developed in Electronics Design Facility at University of North Carolina at Chapel Hill. FSCV system is combined with the equipment for optogenetic experiments via TTL signal.

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
