# Peer review of "Applying a Fast-Scan Cyclic Voltammetry to Explore Dopamine Dynamics in Animal Models of Neuropsychiatric Disorders"

_cells, 2022, doi:10.3390/cells11091533_

Round 1

Reviewer 1 Report

Grinevich et al. review insights into dopamine dynamics in neuropsychiatric and neurodegenerative diseases gleaned from the use of fast-scan cyclic voltammetry (FSCV). A short search of pubmed revealed no similar reviews in the recent literature. Given the utility of the technique, the topic is both timely and appropriate. However, I have several major concerns that bear addressing by the authors:

  1. Given that this review focuses on FSCV as a technique, little space is devoted to describing the technique or listing the advantages/disadvantages of this technique vs others available for monitoring dopamine in vivo (e.g. chronoamperometry, microdialysis, fiber photometry + neurotransmitter sensors). This needs to be included in the introduction. How does FSCV stack up in terms of spatial and temporal resolution, sensitivity, ease of use, and ability to couple to behavior for realtime monitoring?
  2. The authors should expand on their discussion of the FSCV technique or provide additional references. They mention the basis for the technique, but how is it utilized in vivo? Perhaps a model/image of a typical set up for in vivo monitoring would be helpful here.
  3. The first sentence in the introduction is misleading and should be re-written. Mechanisms contributing to these disorders have yet to be fully delineated and this is still an area of ongoing research. Not sure if “major” insights were gleaned 60 years ago??
  4. In lines 84-88 the authors propose that a correlation between DA and behavior following electrical stimulation of VTA neurons does not prove a direct impact of DA. There is a growing body of evidence suggesting that DA neurons release multiple neurotransmitters (e.g. also glutamate) that could be cited here as support for this hypothesis. (see: https://pubmed.ncbi.nlm.nih.gov/31698743/)
  5. In several places throughout the manuscript, the authors employ generic language. For example, line “mechanisms of pathological phenomena”. What does this mean? FSCV is used for a specific application: monitoring DA levels in vivo. The authors need to be more explicit. Other examples:
    • line 48 “other up-to-date approaches and behavioral models”
    • line 268 “the excitatory mechanism”
  6. The addition of a table highlighting the major studies described would be helpful for a reader.
  7. This manuscript contains several errors in English language and would benefit from editing by a native English speaker. This is particularly true in the introduction and conclusion sections.
    • Line 11: “a real time” should be “the real time”
    • Line 12: delete “the” before “researchers”
    • Line 12: insert “the” before “exploration”
    • Line 13: delete “the” before “one”
    • Line 15: delete “the” before “dopamine”
    • Line 16: delete “utilizing”
    • Line 29: replace “No doubts, the success in developing” “The capacity to develop”
    • Line 42: replace “which is in charge” with “which described the”
    • Line 58-59: delete “following treatment cource”
    • Line 71: Replace “was equally proposed as well” with “has also been proposed”
    • Line 76: insert “a” before “subsequent”
    • Line 100: “study” should be “studies”
    • Line 100-101: sentence incomplete
    • Line 113: insert “a” before “phasic”
    • Line 120: “between the patterns of dopamine transmission” and what?
    • Line 120: replace “to find” with “found”
    • Line 140: insert “the” before “consequences”
    • Line 149: “efficacy” should be “affinity”
    • Line 150: “Using self-administration paradigm, which” should be “Using a self-administration paradigm that”
    • Line 152: delete “group of”
    • Line 169: “FSCV ex vivo approach” should be “Ex vivo FSCV”
    • Line 171: “observe” should be “observed”
    • Line 171: “, which” should be “that”
    • Line 178: insert “the” before “dopamine transporter”
    • Line 187: “adaptive alterations due to change in cocaine efficacy”
    • Line 189: “into” should be “in”
    • Line 192: insert “an” before “understudied”
    • Line 192: replace “performed” with “represent”
    • Line 200: “They detected dopamine release in both the…”
    • Sentence on lines 209-210 is incomplete
    • Line 211: “By combining”
    • Line 212: “they explored”
    • Line 216: “at a dose”
    • The sentence on lines 223-225 is confusing and should be clarified
    • Line 225: insert “the” before “evolutionary”
    • Line 243: replace “that” with “whose”
    • Line 247: “onset” should be “appear”
    • Line 284: “provides an intrigued finding such as activity in dopamine cell bodies are” should be “demonstrated that activity in dopamine cell bodies is”
    • Line 290: “to uncover” should be “in uncovering”
    • Line 309: “that considered as model of early PD” should be “, a model of early PD,”
    • Line 312: “dependable” should be “dependent”
    • Line 332: “utilizing the approach mentioned above”
    • Line 349: “the study reviewed above”
    • Lines 349-355 are confusing and should be re-written to clarify
    • Line 363: “allowed for identification of” rather than “allowed to uncover”
    • Line 377: “Also, there numerous modified FSCV techniques have been developed by the present time” should be “Additional modified FSCV techniques have also been developed”
    • Paragraph on lines 389-392 is confusing. It could easily be removed or shortened.
    • Line 398: “which” should be “whose”
    • Line 416: “The dopamine theory is supported by the observation that”
    • Line 422: insert “a” before “hyperdopaminergic”
    • Line 434: insert “a” before “decreased”
    • Line 446: “not directly affecting”
    • Line 458: insert “a” before “second”
    • Line 482: “particularly instrumental in exploring”
    • Line 521: “dependable” should be “dependent”
    • Line 524: “had become FSCV” should be “is FSCV”
    • Line 527: “progressing” should be “progression of”

Author Response

Dear Reviewer,

We appreciate your review report. We agree with all of your major concerns that we addressed at our best effort.

Please note: we were experiencing a slight misplacement of the lines between the “manuscript.docx” file and the review report thus “former lines” designate lines in the latter.

Responses:

  1. As the Reviewer requested, we have expanded the introduction section with more detailed description of the fast-scan cyclic voltammetry in terms of its spatial and temporal resolution, sensitivity and other mentioned features. In addition, we devoted more space for the advantages/disadvantages of the technique vs others available approaches for monitoring dopamine including chronoamperometry, microdialysis, and neurotransmitter sensors with fiber photometry detection (see Introduction section).  
  2. Following the Reviewer’s inquiry, the revised introduction section includes a discussion of the FSCV technique and provides relevant references. As suggested, the image of a typical FSCV set up for in vivo monitoring was included.
  3. We agree with the Reviewer that the first sentence in the introduction was misleading. It has been re-written.
  4. The suggested reference was added (line 167)
  5. We have maximally eliminated generic language, specifically in lines listed below:
  • former line 48 – “other up-to-date approaches and behavioral models” was eliminated
  • former line 268/new 372-374 – “ the excitatory mechanism” was replaced with “activates presynaptic nAChRs, which in turn slightly stimulate dopamine release…”
  1. We appreciate the advice to add a table highlighting the major studies described but we do not see it as an essential for further improvement of our review.
  2. We have corrected the indicated errors in English language as follows:
    • “a real time” replaced with “the real time”
    • “the” was deleted before “researchers”
    • “the” was inserted before “exploration”
    • “the” was delete before “one”
    • “the” was delete before “dopamine”
    • “utilizing” was deleted
    • “No doubts, the success in developing” was replaced with “The capacity to develop”
    • “which is in charge” was replaced with “which pioneered the”
    • “following treatment course” was deleted
    • “was equally proposed as well” was replaced with “has also been proposed”
    • “a” was inserted before “subsequent”
    • “study” was replaced with “studies”
    • specified was completed
    • “a” was inserted before “phasic”
    • “between the patterns of dopamine transmission” – it was clarified
    • “to find” was replaced with “found”
    • “the” was inserted before “consequences”
    • “efficacy” was replaced with “affinity”
    • “Using self-administration paradigm, which” was replaced with “Using a self-administration paradigm that”
    • “group of” was replaced with deleted
    • “FSCV ex vivo approach” was replaced with “Ex vivo FSCV”
    • “observe” was replaced with “observed”
    • “, which” was replaced with “that”
    • “the” was inserted before “dopamine transporter”
    • “adaptive alterations due to change in cocaine efficacy” was used
    • “into” was replaced with “in”
    • “an” was inserted before “understudied”
    • “performed” was replaced with “represent”
    • “They detected dopamine release in both the…” was used instead of passive voice
    • Sentence on former lines 209-210 was completed (new lines 302-304)
    • “By combining” was corrected
    • “they explored” was corrected
    • “at a dose” was corrected
    • The sentence on former lines 223-225 was revised (new lines 315-318)
    • “the” was inserted before “evolutionary”
    • “that” was excluded due to a re-phrasing
    • “onset” was replaced with “appear”
    • “provides an intrigued finding such as activity in dopamine cell bodies are” was replaced with “demonstrated that activity in dopamine cell bodies is”
    • “to uncover” was replaced with “in uncovering”
    • “that considered as model of early PD” is “, a model of early PD,”
    • “dependable” was replaced with “dependent”
    • “utilizing the approach mentioned above” corrected
    • “the study reviewed above” corrected
    • Former lines 349-355 were re-written (new lines 483-490)
    • “allowed to uncover” modified to “allowed for identification of”
    • “Also, there numerous modified FSCV techniques have been developed by the present time” is “Additional modified FSCV techniques have also been developed”
    • Paragraph on former lines 389-392 was removed
    • “which” was replaced with “whose”
    • “The dopamine theory is supported by the observation that” was corrected
    • “a” was inserted before “hyperdopaminergic”
    • “a” was inserted before “decreased”
    • “not directly affecting” – was corrected accordingly
    • “a” was inserted before “second”
    • “particularly instrumental in exploring” – was corrected accordingly
    • “dependable” was replaced with “dependent”
    • “had become FSCV” was replaced with “is FSCV”
    • “progressing” was replaced with “progression of”

Reviewer 2 Report

Dear authors,

“Applying a fast-scan cyclic voltammetry to explore dopamine dynamics in animal models of neuropsychiatric disorders” addresses a topic issue, such as real-time dopamine dynamics in different psychiatric conditions, with special focus to the challenging and undervalued FSCV. I agree that your review would be highly appreciated.

The manuscript It is generally well organized, and citations are appropriate, however writing could be improved. Please find possible suggestions below:

Please check english typing ad grammar, for example: Line 57 and 60 substence → substance; Line 59 Cource → course; Line 398 which → whose (?)

Please check carefully throughout the text for abbreviations already mentioned, for example: Line 174 dopamine transporter (DAT); Line 262 fast-scan cyclic voltammetry (FSCV)

Line 190 to 193 Not clear. Please rephrase.

Line 194 Please specify “excellent spatial resolution” for non expert, for example specifying carbon fiber dimensions as compared with microdialysis probes.

Line 221 to 240. In this paragraph, application of FSCV sounds not directly linked to SUDs. Please specify if this is the case.

Line 242-244 This sentence is not clear and focused enough for introducing PD. Please rephrase.

Line 271-272 Please rephrase

Line 377-381 Please check english. Indeed, novel “modified FSCV techniques” are very attractive in this context and should be more than “just mentioned”. Brief description would be appreciated here or in another section of the manuscript. (This is well done by authors about M-CSWV in lines 383-387).

Line 398-399. Please provide at list a short presentation of featuring symptoms in Schizophrenia before molecular mechanisms. Mere citation of Kahn et al. sounds not enough and non-uniform with previously presented pathologies.

Conclusions could be enriched with "future directions" (such as the above mentioned "modified FSCV techniques”) or mentioning other psychiatric disorders that could benefit from FSCV, such as ADHD.

Author Response

Dear Reviewer,

We appreciate your review report. We agree with all of your major concerns that we addressed at our best effort.

Please note: we were experiencing a slight misplacement of the lines between the “manuscript.docx” file and the review report thus “former lines” designate lines in the latter.

Responses:

  • By the Reviewer’s request, we checked English typing and grammar and corrected accordingly
  • All abbreviations, DAT and FSCV in particular, were checked and corrected
  • The phrase in former lines 190 to 193 was re-written (new lines 280-281)
  • As the Reviewer suggested, we clarified excellent spatial resolution of FSCV for non-experts
  • As the Reviewer requested, we made application of FSCV sounded linked to SUDs
  • The sentence in former line 242-244 was corrected accordingly (new lines 337 -340)
  • The sentence in former line 271-272 was removed since entire paragraph was re-written (new lines 358-380)
  • As the Reviewer suggested, we provided more details on modified FSCV variations in Introduction and Conclusion sections
  • To follow up the Reviewer’s request, we added a brief description of featuring symptoms of Schizophrenia and the citation was used appropriately
  • We enriched Conclusion section with future directions in the field of FSCV and added discussion on other psychiatric disorders that could benefit from voltammetry studies

Round 2

Reviewer 1 Report

The authors have addressed my comments satisfactorily. There are still some grammatical errors that should be correct in proof. 

Reviewer 2 Report

I appreciated Author's revision to the manuscript and consider the Review in the present form as

useful for the field